# Effectiveness of Nutritional Strategies on Improving the Quality of Diet of Children from 6 to 12 Years Old: A Systematic Review

**DOI:** 10.3390/nu14020372

**Published:** 2022-01-15

**Authors:** Naroa Andueza, Santiago Navas-Carretero, Marta Cuervo

**Affiliations:** 1Department of Nutrition, Food Sciences and Physiology, School of Pharmacy and Nutrition, University of Navarra, 31008 Pamplona, Spain; nandueza@unav.es (N.A.); mcuervo@unav.es (M.C.); 2Center for Nutrition Research, School of Pharmacy and Nutrition, University of Navarra, 31008 Pamplona, Spain; 3Biomedical Research Networking Center for Physiopathology of Obesity and Nutrition (CIBERObn), Institute of Health Carlos III, 28029 Madrid, Spain; 4Navarra Institute for Health Research (IdiSNA), 31008 Pamplona, Spain

**Keywords:** diet quality, dietary pattern, children, nutritional strategy

## Abstract

Dietary habits, that are formed during childhood and consolidated in adulthood, are known to influence the development of future chronic diseases such as metabolic syndrome or type 2 diabetes. The aim of this review was to evaluate the effectiveness of nutritional interventions carried out in recent years focused on improving the quality of the diet of the child population. A systematic search of the PubMed and Scopus databases was performed from January 2011 until September 2021. A total of 910 articles were identified and screened based on their title, abstract and full text. Finally, 12 articles were included in the current systematic review. Of those, in six studies the intervention was based on the provision of healthy meals and in the other six studies the intervention focused on modifying the school environment. Six of the studies selected included other components in their intervention such as nutritional education sessions, physical activity and/or families. A wide variety of methods were used for diet assessments, from direct method to questionnaires. The results suggest that interventions that modify the school environment or provide different meals or snacks may be effective in improving children’s dietary patterns, both in the short and long term. Further research is necessary to evaluate the real effectiveness of strategies with multidisciplinary approach (nutritional sessions, physical activity and family’s involvement).

## 1. Introduction

Dietary habits are formed during childhood and consolidated in adulthood. They may also have influence in the development of future chronic diseases such as metabolic syndrome, type 2 diabetes, cardiovascular disease and mortality [1,2,3]. Therefore, different institutions, including the World Health Organization, recommend the establishment of healthy eating habits at an early age as a method to prevent chronic diseases [1,2,3,4]. Moreover, poor dietary habits are associated with being overweight and obesity, including comorbidities such as fatty liver disease, dyslipidemia, diabetes, asthma, sleep apnea and cardiovascular disease among others [5]. In addition to health consequences, childhood obesity can affect children’s social and emotional well-being and self-esteem [5] as well as educational achievement [6]. Furthermore, it is associated with a lower quality of life [5] and with a higher risk of obesity in adulthood [7].

However, it has been observed that the majority of children of Western populations do not achieve the goals of dietary guidelines, as they consume low amounts of fish, fruit, vegetables and foods high in fiber, and large amounts of foods high in sugar and saturated foods [8].

Diet-related risk factors are a major cause of death and disease worldwide. In the case of children, it has been observed that a high intake of sodium, saturated fat, meat, fast food and soft drinks is negatively associated with cardiovascular health while a high intake of vitamin D, fiber, mono- and poly-unsaturated fatty acids, dairy products, fruits and vegetables is positively associated [9]. Additionally, it has been observed that a high intake of sugar, and particularly the intake of sugar-sweetened beverages and added sugar, is a risk factor for the development of obesity, dyslipidemia, type 2 diabetes mellitus [10] and caries [11], among others.

In addition, there are certain risk eating behaviors associated with different diseases. For example, it has been observed in children and adolescents that skipping breakfast is a risk factor for obesity [12] and metabolic diseases [13]. It is associated with a worse lipid profile, blood pressure levels and insulin resistance too [13].

Changes in lifestyle of young people around the world have been observed in the last decades, leading to changes in their dietary patterns. These changes, in part, appear to be due to the impact of globalization and urbanization on eating patterns. A greater trend has been observed for the consumption of snacks, meals away from home, fast foods and sugar-sweetened beverages. In short, there is a trend towards the consumption of highly processed foods with low nutritional density [14,15,16]. In addition, it has been observed that adolescents from different regions of the world (North America, Europe and Oceania) do not follow the nutritional recommendations for fruits, vegetables, legumes and sodium, nor do they follow a Mediterranean diet pattern [16]. For this reason and given that eating habits are established in childhood, it is really important that before reaching these stages, adolescence and adulthood, an adequate eating pattern is consolidated.

Dietary patterns have been shown to be strongly influenced by family environment [3] and food availability [3,17]. First, regarding the family environment, it has been observed that the main factors that affect children’s eating behaviors are: availability and accessibility of healthy foods, frequency of family meals, parental intake and parenting practices [18,19]. Regarding the availability and accessibility of healthy foods, it has been observed that the intake of fruits and vegetables increases when these two food groups are available at home even when the taste preferences towards them is low [18]. It has also been observed that one of the key elements in increasing the intake of soft drinks is their availability at home together with taste preference [20]. Parental intake and frequency of meals is another essential element. In this context, parents play a fundamental role in the development of children’s eating habits because children tend to modulate their eating behavior based on their parent’s habits [21]. Besides, having a family dinner is related with higher quality diets, characterized, among others, by a higher consumption of fruits and vegetables and a lower consumption of fried foods, soft drinks and food high in saturated and trans fats [22]. With regards to parenting care practices, it has been observed that children living in authoritative homes eat healthier, are more physically active and have a lower BMI compared to children who were raised in other types of homes such as authoritarian, permissive/indulgent or not involved/neglectful [23]. Moreover, family eating patterns, healthy eating rules at home, and parents’ healthy lifestyles have been shown to influence young people’s intake of fruits, vegetables, calcium and dairy products and dietary fats [24]. Moreover, it has been stated that those nutritional strategies involving parents in the promotion of healthy eating in children improves the quality of children’s diets as well as reducing the prevalence of childhood obesity [21].

On the other hand, regarding the availability of food, it is known that neighborhoods that offer access to a high nutritional quality food improve the diet and weight of the people who live in those neighborhoods. In that sense, it has been shown that access to foods and beverages with a high energy content has negative consequences on the health of people around the world. In addition, the increase in the price of fruits and vegetables in contrast to that of processed products has made the access to these not-so-healthy products very easy. Given this trend in food prices, the population group that has been most affected are those with low incomes [17]. Therefore, to achieve the improvement of the population’s dietary and lifestyle patterns, it is necessary that healthy foods are available, identifiable and affordable for all, regardless of the income level and the area of residence (rural or urban) [18].

In addition, schools are one of the places where children spend most of the day [25,26]. A large part of this population has one of their main meals, lunch, there. Approximately 40% of the total energy intake is consumed in schools [27,28], where there are other sources of food and drinks as well, apart from the cafeteria service, such as vending machines, school stores or food stands [29]. Generally, the type of food coming through these channels is low in nutrients and dense in energy [28]. On the other hand, schools are places where children are exposed to physical exercise, where many of them provide health education and a healthy environment and allows access to a large number of children [26]. In this context, these centers can promote healthy eating by increasing the availability or limiting different foods based on their nutritional quality and teachers can promote healthy eating through nutrition education in the classroom [29]. Therefore, schools are key places for the development of strategies focused on improving eating habits [25,26]. A systematic review suggested that school food and nutrition policies (nutrition guidelines, healthy food price interventions, and fruit and vegetable subscription or distribution plans) might be effective in improving the school feeding environment and the dietary intake of children. However, these strategies were not effective on body weight [30]. Nevertheless, the current scientific evidence on the effectiveness of this type of strategy is still limited [31].

Nowadays, scientific evidence on interventions that improve the nutritional quality of the child population remains limited. No agreement has been reached on the characteristics of the intervention that is responsible for said effectiveness. The reviews on this topic have focused exclusively on the school environment or are not recent. Additionally, these papers are not consistent in their results [32,33,34]. The present review intends to increase the scientific evidence on this topic and provide updated data, since the problem of low-quality diets in children is a frequent health problem. Therefore, interventions that are truly effective in the proposed population could be developed and implemented.

The aim of the present review was to evaluate the efficacy of different types of interventions carried out in children between 6 and 12 years old focused on the improvement of the quality of the diet in recent years. As secondary objective, to identify and analyze the essential components involved in the change of the nutritional quality of the diet, the magnitude and impact of the results obtained on the diet and the other possible effects that could derive from this improvement on the diet.

## 2. Materials and Methods

### 2.1. Search Strategy

A bibliographic search was carried out in two online databases PubMed and Scopus databases from the January 2011 until September 2021. The following groups of keywords were used for the research: (1) “nutrition* intervention” AND “diet* quality” AND “(school meal OR canteen OR food)” and “child”—(2) “lifestyle* intervention” AND “diet* quality” AND “(school meal OR canteen OR food)” AND “child”—(3) “nutrition* intervention” AND “school meal” AND “diet*” AND “improve” AND “child”. To appropriately conduct the article search, the following filters were applied in the databases: year (last 10 years), language (English) and type of article/document (Pubmed: clinical trial; Scopus; article).

Articles published in English, carried out in children and that looked for associations between different types of interventions in which one of its components was nutritional and focused on diet/lunch/school meals and the improvement of the quality of the diet or dietary pattern, which in addition were clinical trials or used a similar approach, were selected.

The research was carried out using the PICO strategy (patient, intervention, comparative, results). The question according to the PICO scheme was: “In children, what type of intervention is found to be effective improving the quality of the diet or the dietary pattern compared to control?”

### 2.2. Inclusion and Exclusion Criteria

All studies carried out in children could be included. Taking into account that many of the characteristics of both the nutritional status and the diet of children change significantly throughout growth, it was decided that a study stage would be established between 6 and 12 years. Those studies whose age of participants was within this range were included. The population that is the object of the current review is the child population. Therefore, all articles studying results on children who met the established age range were selected, regardless of their nutritional status (normal nourished, malnourished or obese). Only children from special education schools were excluded because these children usually present pathologies that require specific nutritional care. Furthermore, all the included studies had to evaluate as one of their results the quality of the diet or dietary pattern (primary outcome of this review), thus being able to determine whether a change occurred after the intervention. Articles involving different intervention approaches were included, both isolated nutritional interventions and those that, in addition to the nutritional component, presented other components such as nutrition education and/or physical activity in their intervention. This second type of intervention was named by the authors as lifestyle interventions. Regarding the studies design only prospective controlled studies were selected.

### 2.3. Data Extraction

Data were extracted in a standard way. For each included study, the following information was obtained: title, author name, year of publication, study design, subjects’ characteristics, sample size, dietary interventions’ characteristics, diet assessment methods and main results.

### 2.4. Search Summary

A total of 910 articles were identified from two electronic databases: PubMed and Scopus. Of all these articles identified, 266 were duplicates, so they were removed, resulting in a total of 644 articles. After a screening based on the title, 97 articles were eligible for abstract screening, of which 30 articles were qualified for a full-text review. After complete revision of the text, 18 articles were eliminated for not meeting the inclusion and exclusion criteria mentioned above, such as the age of the participants, that a nutritional intervention will be carried out on diet, lunch or school meals and that one of the results measured the improvement in the quality of the diet or dietary pattern thus assessing the change after the intervention. In addition, two other articles were identified through articles from the previous search. Finally, 12 articles were selected for the final review. The present review followed the PRISMA guidelines and Figure 1 shows the PRISMA Flowchart of the study [35,36].

## 3. Results

The main characteristics of the included studies are described in Table 1. From a geographical point of view, four studies were conducted in Europe [37,38,39,40], six in the United States [41,42,43,44,45,46], one in China [47] and another in Australia [48]. In relation to the study design, ten were randomized controlled trials [37,39,41,42,43,44,45,46,47,48], one non-randomized trial [40] and one longitudinal quasi-experimental trial [38].

Regarding the sample size of the studies, it ranged from less than 100 to more than 10,000. In this context, one study presented a sample size of <100 participants [41], five studies had a sample size between 100 and 1000 [37,40,42,44,45], another five had a sample size between 1000 and 5000 [38,39,43,46,47] and another one had a sample size greater than >10,000 participants [48].

As established in the inclusion and exclusion criteria, the target population for this review was children aged between 6 and 12 years. In this context, the age of the study participants met this age range, although some studies established 5 years old as the lower eligible age, while the upper range could be up to 15 years old.

In relation to the intervention type, six studies were nutrition interventions [37,39,40,41,43,48] and the other six were lifestyle interventions [38,42,44,45,46,47]. Regarding the nutritional component of all the interventions, there were two main models: provision of healthy meals during school hours [37,38,39,40,41,47] and modifying the children’s food environment [42,43,44,45,46,48]. In interventions in which a healthy meal was provided, there were differences regarding the meal intake; one study focused on breakfast [39], two on lunch [40,47], another two on lunch and mid-morning snack [37,38] and one that did not focus on any particular intake but rather provided two snacks to be consumed throughout the day [41]. In interventions in which children’s food environments were modified, in four of them the intervention was focused on modifying the school environment [42,43,48,49], while the two other interventions stressed the change outside school, such as community recreation centers, corner stores and carryout restaurants [45] or after-school sites [44]. In the case of the four strategies based on schools, in two of them the modification of the school environment was based only on the modification of the school food service and in the remaining two, in addition to that change, other actions were carried out aiming at the promotion of health such as: changes in the school physical education program [46], modification of the type of food offered by vending machines [46] and the incorporation into the school curriculum of training activities on healthy eating and physical activity [42]. In the case of the two studies that focused their interventions on modifying the environment outside of school, apart from increasing the availability of healthier food in different retailers, healthy eating habits were also promoted through social networks [45], text messages, posters and handouts promoting healthy food items were placed in all intervention stores [45] and brochures with nutritional messages for families [44].

In addition, six of the studies included other components in their intervention, such as sessions of healthy nutrition and habits or physical activity. In the case of the sessions about healthy nutrition and habits, these were aimed at different groups: children participating in the study [45], children and their families [47] or the school community [42,44]. With regards to physical activity, although in some of the studies physical activity was promoted as part of the healthy habits sessions, in two of the studies physical activity sessions were included as part of the intervention, either daily or a couple of times a week [38,47].

The duration of the intervention period varied depending on the study: the shortest period was two months [41] and the longest was two years [38]. The majority of the studies, eight of the twelve selected, had a duration equal to or greater than one academic year [38,39,40,42,44,45,46,47,48]. All studies performed final measurements at the end of the intervention, while none of the studies performed longer-term follow-up measurements.

Different diet assessment methods were used to collect food information: in four studies direct observation methods were used to determine the type and quantity of food consumed [43,44,46,48] and in the remaining eight studies different questionnaires were used [37,38,39,40,41,42,45,47]. Different types of direct observation methods were used: digital photography of the food and drink selected and consumed [46], observation and recording by trained staff of the type and quantity of food and drink selected and consumed [43], direct observation only of the food selected without assessing the real intake made [48] or a combination of both methods, observation and recording by the study staff plus digital photography [44].

Out of the 14 questionnaires used to assess diet among all studies, excluding the two questionnaires that only recorded the consumption of food provided in the study [38,41], eight of them were questionnaires validated in the target population (children) [37,39,40,41,42,47], one was a validated questionnaire but not in the target population (validated in adolescents) [45] and three were not validated questionnaires [38,39,41].

In relation to questionnaires’ type there were: three 24 h dietary recall [39,41,42], one food record of the last seven days [37], three food frequency questionnaires [40,45,47], one questionnaire that focused only on the consumption of fruits and vegetables [47], another that raised general questions about the intake of different food and beverage groups and dietary behavior [38], two questionnaires about dietary behavior [39,45] and one about gastrointestinal health [41].

With regards to the main result of the studies, expressed as change in the quality of the diet or the dietary pattern, in ten studies an improvement of diet quality was observed after the intervention [37,38,39,40,42,43,44,45,47,48], while in two studies no change in the participants’ diet quality was observed after the intervention [41,46]. Regarding those studies in which improvements on the dietary pattern were observed, in all of them the improvement was mainly focused on an increased consumption of foods considered as healthy (fruits, vegetables, cereals, dairy products, fish, white meats and water) and a decrease in those considered as unhealthy (pastries, processed meats, commercial juices, sugary drinks and unhealthy snacks such as chips). One of the common improvement factors among the studies was a higher consumption of fruits and vegetables. Six studies reported a higher intake of fruits and vegetables: three studies only observed improvements in the consumption of vegetables [37,38,40] and the remaining three studies in the consumption of both groups, fruits and vegetables [42,43,47]. In another two studies, a higher consumption of foods considered healthy was observed, among those that included fruits and vegetables in that classification; however in these results, it was not differentiated which specific food groups had increased their intake [39,48]. Another factor among the studies is the generalized decrease in the consumption of juices and sugary drinks [37,43,44,48].

In relation to dietary macronutrient distribution, an improvement was observed in four studies [37,43,44,48]: in two of them a decrease in fat intake was observed [37,48] and in the remaining two a decrease in total calorie intake [43,44]. Additionally, in the study carried out by Andersen et al., 2014 [37], these other effects were observed: increase in protein intake, increase in dietary fiber consumption that borderline statistical significance and changes in micronutrient content observing an increase in the intake of vitamin D and iodine. In the remaining studies, no nutritional improvement in the diet was observed.

Changes in physical activity and sedentary attitudes were observed in two of the studies. In the study conducted by Li et al. 2019 [47] a higher proportion of children who carried out some type of activity during the weekend and a lower proportion of children with sedentary behaviors were observed at the end of the intervention. Similar to these results, the study by Bartelink et al., 2019 [38] observed a decrease in the percentage of time that children spent sedentary and an increase in the time they spent in moderate activity.

Furthermore, in some studies other secondary results were observed such as: a decrease in BMI z-score and waist circumference [47], lower glycemic index of the diet [42], better punctuation on the quality of life questionnaire [47] or lower energy density of food and beverages [37]

There was a lack of information on the process of implementation of the study and evaluation of the intervention [37,38,39,41,42,43,44,45,46]. Some of the studies also did not specify the study design in detail, such as study enrollment rate [41,42,43], the baseline dietary characteristics of the participants [43,46], or the attrition rate [38,40,43]. These factors present a possible risk of bias in the studies, but in general most of them reported all the study data, so the evidence from the reviewed studies is strong.

## 4. Discussion

This review tried to summarize the evidence on the effectiveness of different interventions focused on improving the quality of the children’s diets in order to be able to develop future interventions that produce a real and long-term impact on the diet of the child population. Of the evaluated studies, ten out of twelve, showed positive associations between specific interventions and an improvement on diet [37,38,39,40,42,43,44,45,47,48].

In those studies, in which improvements were observed on the dietary pattern, one of the common improvement factors among the studies was a higher consumption of fruits and vegetables [37,38,40,42,43,47]. The low intake of these two food groups is one of the most worrisome in this target population (children) as it can be particularly difficult to change their consumption. Despite this, Evans et al., 2012 [50] in their review of school interventions focused on increasing the consumption of fruits and vegetables, concluded that these interventions were successful in increasing the consumption of fruits, but not vegetables. Therefore, the results obtained in this review are encouraging and show that there are nutritional strategies capable of increasing the consumption of one of the food groups that are least accepted in childhood, vegetables. Another common improvement factor between studies was a decrease in the consumption of juices and sugary drinks [37,43,44,48]. This was another of the key dietary factors in children associated with multiple diseases and whose intake had increased in recent times [51,52].

Improvements in the macronutrient content of the diet were also reported in four of the reviewed studies [37,43,44,48]. Given that most of the selected studies observed changes in dietary intake tending towards a healthier eating profile, it is foreseeable that in these cases, improvements in caloric and macronutrient content would occur.

In only one of the reviewed studies, changes were observed in the anthropometric variables. The study by Li et al. 2019 [47] reported a significant reduction in the BMI z-score and in waist circumference after one year of intervention. In this context, Alman et al., 2015 [53] observed that the decrease in meals away from home was associated with better diet quality and a reduction in BMI and body fat percentage, and that the relationship between the decrease in the number of meals away from home and the improvement in anthropometric variables was mediated by the improvement in diet quality. This statement is also supported by the results obtained in the study carried out by Williamson et al., 2012 [46] in which no change was observed in the dietary pattern and therefore in the anthropometric variables. Consequently, it is possible that if the rest of the studies anthropometric and body composition measurements had been taken, changes in the dietary pattern would have led to improvements in these variables.

Moreover, two studies found an improvement in physical activity and reduction in sedentary behaviors [38,47]. This emphasizes the importance of carrying out interventions that do not focus only on improving a specific healthy habit, but rather the development of nutritional strategies focused on the acquisition of a healthy lifestyle as a whole, which includes physical activity. In only these two studies, one of the components of the intervention consisted of physical activity sessions [38,47]. Related to this, recent investigation suggests that dealing with two health behaviors grouped together could produce an indirect or synergistic effect, whereby the probability of improving one health behavior increases when an individual has already successfully changed the other health behavior [38]. Therefore, strategies with this multidisciplinary approach (nutrition and physical activity) might be more effective.

However, two of the studies did not show any effect on dietary pattern [41,46]. First, in the study carried out by Brauchla et al., 2019 [41], in which two fiber-rich snacks were provided to the participants in the intervention group, the authors pointed out that the participants in that group did not consume snacks usually, so despite no major dietary changes, the children were probably replacing refined grains with whole grains, which is a positive dietary change. This must be added the small sample size, less than 100 participants. Second, in the study carried out by Williamson et al., 2012 [46], there was a low rate of implementation of these strategies, a modification program of the school environment, so this could explain that no changes would have observed.

An interesting point to highlight is one of the results of the study conducted by Murphy et al. 2011 [39]. In this study, despite having observed that the nutritional strategy impacts on one of the meals of the day, producing a greater consumption of healthy foods at breakfast, no effect was observed on the rest of the meals of the day. These results would have a greater impact and benefit for the participants if they were able to produce improvements in the overall dietary pattern.

With regards to the main characteristics of the studies, most of them had large sample sizes, specifically ten of the selected studies involved more than 400 participants, which allows reliability in the results obtained. However, there were large differences in the sample size of the studies. Perhaps the small sample size of the remaining two studies is due to the fact that the study was carried out in rural areas where there is generally a smaller population [39,40]. In the case of the studies carried out in schools, it is also important to note that although some studies had a high number of participating schools, in some cases the number of children enrolled in the study was really low, as is the case of the studies carried out by Cohen et al., 2014 [42] and Wolfenden et al., 2017 [48]. Thus, these results should be interpreted with caution.

Another of the main characteristics is the duration of the intervention period. In general, the selected studies presented long intervention periods (in seven of them it was at least one academic year), which suggests that these strategies could be effective in the long term. This would be the case of the studies carried out by Wolfenden et al., 2017 [48] and Bartelink et al., 2019 [38] whose duration is one and two years, respectively. Given that these studies are carried out in the school environment, it is to be assumed that at least during the summer vacation period the intervention was not continued.

In this context, it is necessary to emphasize that the effects obtained in the interventions tend to fade over time [54,55,56]. It would have been interesting if the selected studies had evaluated these long-term effects after the intervention was completed, for example during the following academic year. In this sense, it could be evaluated if the new healthy habits acquired are maintained over time and especially once the intervention is finished, since this is the ultimate objective of any study. A recent study carried out in an adolescent population that evaluated the long-term effect of a school-based intervention on the nutrition and physical activity on BMI, found that after two years of the intervention it was still being effective, mainly in obese participants [57]. Similar results were observed in Mihas et al., 2010 nutritional intervention study on BMI one year post-intervention [58]. In both studies, no dietary variables were determined, but since improvements in weight were observed, it is more than likely that there were also improvements in diet.

Regarding the type of intervention carried out, most of the included studies focus on schools or on children’s environments, but neither of them focused on both areas. This would be another point to be developed in future interventions in children in order to determine if developing an intervention based on both the school and children’s environments produces greater effects on diet and on other series of concomitant variables such as anthropometric and body composition. It would be interesting to assess, for example, the effect of interventions that not only modify the school food service but also the rest of the food services in the environment such as recreation centers, restaurants and/or supermarkets, so that access to healthy food is facilitated in all the areas in which the child relates.

Six of the selected studies counted the provision of healthy and free food as one of the main elements of their intervention [37,38,39,40,41,47]. Facilitating access to this type of food can be key to the success of this type of intervention. In this regard, you do not limit yourself to facilitating the availability of this type of food (as in the case of the modification program of the food served in the school cafeteria) but you directly give them access to them. Thus, children can try foods that they normally do not consume and gradually change their food preferences towards a healthier profile. Different authors support this statement that the fact that children have access to free food not only improves the nutritional quality of their diet, especially in low-income families, but also represents a unique opportunity to modify food preferences, by facilitating access to new foods [59,60].

Related with that, school meal programs have proven to be effective in improving the nutritional quality of children’s diets [61]. Likewise, recent studies suggest that students who bring lunches from home have poorer nutrition compared to those students who consume the foods provided in school meal programs [62]. This highlights once again the importance of schools and specifically school food services in nutritional interventions carried out in the child population. The data seem to suggest that both the provision of food and the modification of food proportions in school lunch services are an effective measure to improve the quality of the diet.

There were two different types of interventions in the selected studies: nutritional interventions and lifestyle interventions. The data suggest that lifestyle interventions are more effective by encompassing more aspects and not focusing solely on nutrition. As mentioned above, focusing on more than one health habit results in them enhancing their effects on each other. A recent review on the effect of interventions for the prevention of obesity in children and adolescents concluded the same, that interventions that combine diet and exercise are more effective [63].

Only three of the studies took into account families in their intervention, developing different activities or educational sessions for parents with the aim of promoting a healthy diet and lifestyle [42,44,47]. In a recent systematic review on the effectiveness of family-based nutritional interventions in improving the diet of children, it was concluded that these types of strategies have a high potential in improving the quality of the diet, mainly being effective in reducing dietary fat and increasing the consumption of fruits and vegetables [64].

There was no consistency in the method for diet assessment. These ranged from direct methods (observation and photography) [43,44,46,48] to different questionnaires filled out by participants and/or parents [37,38,39,40,41,42,45,47]. Direct methods have demonstrated to be reliable and accurate in school studies [49,65]. There are validated methods in the child population of all types of questionnaires used. There is no consensus on which method should be used to evaluate children’s diets, there are different types and there are validated models of all of them. It is true that the use of questionnaires presents limitations, since the data obtained may be biased among others. However, it has been shown that school-aged children are able to respond to data about themselves such as those related to health, as early as six years old [66]. Besides, in three studies questionnaires to assess eating behavior of children were included, which is useful to assess, among others, the attitude children have towards food [38,39,45]. Perhaps, a good way to evaluate diet globally is to use a combination of both methods. In future studies, the possibility of incorporating diet scores into the evaluation methods should be considered, such as the Kidmed Index, which is a quick and validated method to assess the quality of a child’s diet [67].

None of the selected studies considered biochemical variables such as lipid or glycemic profile. In a 2013 systematic review on the effect of nutritional and physical activity interventions on weight and metabolic outcomes in obese children and adolescents, the authors concluded that these strategies were effective on these metabolic variables, especially on HDL cholesterol and fasting insulin levels [68]. Thus, it would have been interesting if some of the selected studies had included biochemical parameters of the lipid and glycemic profiles of the children after improving the nutritional quality of their diets throughout the intervention study. It is possible that improvements had been observed in some of the parameters analyzed, especially in those studies with longer intervention periods.

The main risk of bias in these types of studies is that although these types of studies usually have large sample sizes, the families that participate are often concerned about the health of their children. Thus, in some cases it can be difficult to reach the population that would really benefit from these interventions. For this reason, it is important to evaluate the baseline characteristics of the participants because participating children might generally follow a good quality diet since their parents are very concerned about their diet, so the change produced after the end of the study may not be as big as expected.

At last, there are several limitations of this systematic review. Firstly, not all of the selected studies are randomized controlled trials. Second, most of the result are based on self-reported data. The present systematic review also has strengths. First, it presents a comprehensive and updated evaluation of this topic, analyzing and evaluating each of the components that can make these strategies effective. Second, the large sample size and duration of the intervention of the studies selected shows that the data are reliable.

## 5. Conclusions

The results of the present review suggest that interventions that modify children’s environments or provide different meals or snacks are effective in improving the dietary pattern of the child population in the short and long term. However, it would be interesting to develop interventions that include all the following components in order to produce a greater impact on the diet and health of the child population: (1) strategies that not only focus on improving the diet, but on the acquisition of a healthy lifestyle as a whole, thus incorporating physical activity into the intervention (2) strategies that focused on school and on children’s environments which includes, among others, involving parents in the intervention, due to the important role they play in the diet and habits of children (3) strategies that incorporate nutritional education such as healthy eating sessions (4) strategies that evaluate long-lasting effects once the intervention is finished to be able to determine whether the beneficial results obtained are maintained over time. On the other hand, it would also be interesting for these strategies to evaluate other possible effects such as improvements in weight and body composition or in biochemical markers.

## Figures and Tables

**Figure 1 nutrients-14-00372-f001:**
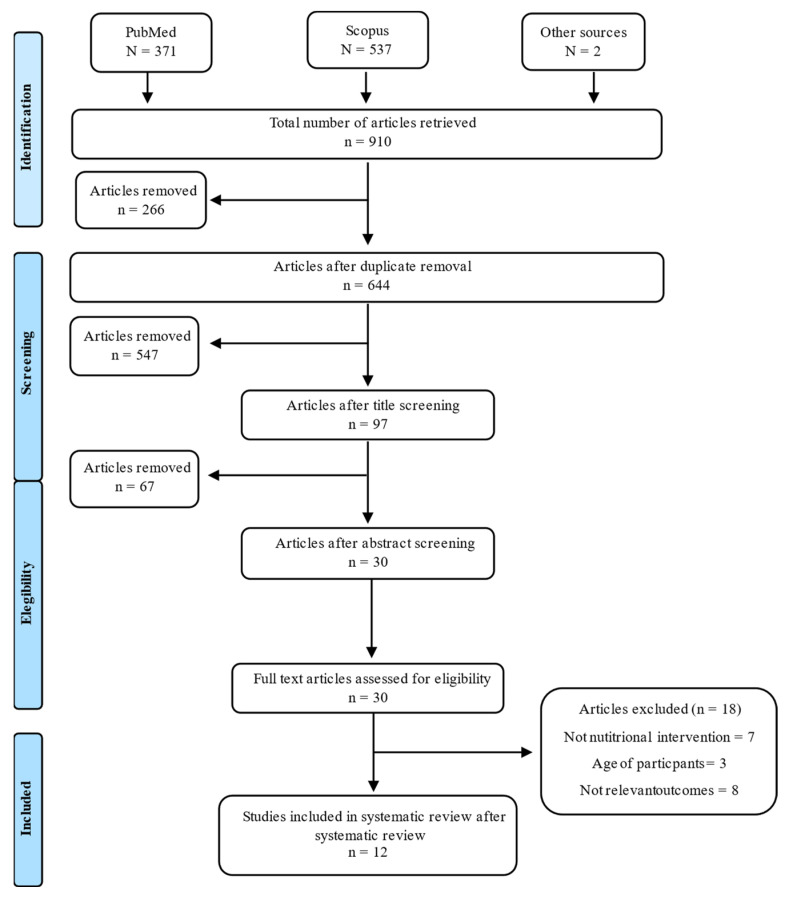
Flow diagram of identification, screening and selection process for included articles.

**Table 1 nutrients-14-00372-t001:** Characteristics of the studies included in the systematic review.

Reference	Study Design	Sample Size	Age	Intervention Description	Diet Assessment	Results
Murphy et al., 2011 [39]	Cluster, randomized controlled trial 1 year. PSFBI	4472	9–11 years	Free school breakfasts containing milk-based drinks or products, cereal (not sugar coated), fruit and breads compared with their usual breakfasts.	Dietary recall questionnaire.Questionnaire about child’s dietary behavior.	Higher consumption of healthy food items in intervention group than control group. Intervention group had more positive attitudes towards breakfast eating than control group.
Williamson et al., 2012 [46]	Cluster, randomized 3-arm controlled trial. 3 academic years. LA Health	2021	7–12 years	(1) Primary Prevention (PP): school environment modification including changes in cafeteria food service and physical education program. (2) Primary + Secondary Prevention (PP + SP): added a classroom/Internet component. (3) Control (C).	Digital photography of food selections and food intake.	No differences between PP + SP and PP and C were found for changes in BMI, body fat, food intake, physical activity or sedentary behavior. Both intervention groups combined (PP + (PP + SP)), were associated with a decreased in body fat and dietary fat intake in comparison to control group.
Brauchla et al., 2013 [41]	Cluster, randomized controlled trial. 2 months.	81	7–11 years	Two high-fiber snacks per day total of 10–12 g of dietary fiber compared with their usual snacks.	24 h dietary recalls via telephone. 8-question Child Regularity Questionnaire. Snack frequency questionnaire.	No differences were observed in terms of energy content, macronutrients, fiber, food groups and punctuation of Regularity Questionnaire between both groups.
Andersen et al., 2014 [37]	Cluster-randomized, controlled, unblinded, cross-over trial. 3 months. OPUS School Meal	834	8–11 years	Free healthy school meals: mid-morning snack, an ad libitum hot lunch meal and afternoon snack compared with their usual packed lunches.	WebDASC: food record tool.	Higher intake of potatoes, fish, cheese, vegetables, eggs and drinks and a lower intake of bread and fats during intervention period compared to control period. Lower energy density of food and beverages during intervention period than in control period. No differences were observed in energy intake during intervention period compared to control period. Higher energy intake from protein (0.9%) and lower energy intake from fats (0.9%) in intervention period. Higher intake of vitamin D (42%) and iodine (11%) in intervention period.
Li et al., 2019 [47]	Parallel, two-arm, cluster-randomized controlled trial. 1 year. CHIRPY DRAGON	1641	6–7 years	In the intervention school the following components were carried out while controls schools continued with usual practice: Education workshops, healthy behavioral challenges and quizzes for families and children. Provision of school lunch Family friendly games at school and home. 1-h physical activity on campus every day.	Short form of SFFQ from University of Leeds. Day in Life Questionnaire.	Higher daily intake of fruit and vegetables and proportion of children consuming at least 5 daily portions of fruit and vegetables in intervention group than in control group. Lower weekly consumption of sugar-sweetened beverages and unhealthy snacks in intervention group compared to control group. Lower proportion of children with screen-based sedentary behavior and higher proportion of children engaging active sport in intervention group compared to control group. More favorable score in questionnaire of quality of life in intervention group than in control group. Decrease in BMI z-score and waist circumference in intervention group compared to control
Cohen et al., 2014 [42]	Randomized, controlled trial. 1 academic year CHANGE	432	6–12 years	Daily access to a food service offering healthier school breakfasts and lunches. Healthy habits acquisition curricula. Different components to promote healthy lifestyle changes for parents and community encouraged during and after the school day.	Block Food Screener	Higher consumption of vegetables and fruits and vegetables combined in intervention schools compared to control schools. No difference was observed in fruit, legume, whole grain or dairy intake. Lower glycemic index of the diet in intervention schools compared to control schools.
Cullen et al., 2015 [43]	Randomized, controlled trial. 6 months.	1876	5–14 years	Intervention and control schools served the same menu, with the difference that intervention group could select one fruit and two vegetables servings per day and control group could only select a total of two servings of fruits and/or vegetables.	Direct observation in the cafeterias during lunch periods.	In elementary intervention schools, increase in the consumption of total vegetables, starchy vegetables and other vegetables and decrease in the consumption of calories juice, whole grains and protein foods compared to control schools. In intermediate intervention schools, increase in the consumption of fruit, total vegetables, starchy vegetables, legumes and decrease in the consumption of calories, total grains and whole grains compared to control schools.
Wolfenden et al., 2017 [48]	Randomized controlled trial. 1 year	57 schools (mean number of students: 256 in intervention group and 253 in control group)	5–12 years	Implementation of a healthy canteen policy that required schools to eliminate unhealthy items from the regular sale and increase those healthy ones.	Direct observations of mean energy, total fat and sodium per student purchase were assessed during one school day.	Intervention schools were more likely to have menus without unhealthy items and to have at least 50% of menu items classified as healthy than control schools. Student purchases from intervention school canteens were lower in total fat but not in energy or sodium than control schools.
Lee et al., 2018 [44]	Group-randomized controlled trial. 1 academic year. OSNAP	400	Mean age: 7–6 years in control group and 7–8 years in intervention group.	Implementation of menus that increased the frequency of fruits; reduced the frequency and servings of juice; removed foods with partially hydrogenated oils; and included more whole grain foods. Healthy habits promotion and nutrition sessions for directors, support staff, families and children	Direct observation and digital photography of type, size and brand of all food and beverage items served each day.	Decrease in the consumption of juice, beverage calories, foods with trans fats, total calories and increase in the consumption of whole grain in intervention group than in control group.
Trude et al., 2018 [45]	Group-randomized controlled trial. 8 months. BHCK	509	9–15 years	Environment modification program and increase in the availability of healthy beverages, snack and cooking methods. Nutrition sessions for children. Promotion of healthy eating through social networks.	CIQ: tool for measuring youth purchasing behavior. BKFFQ: to collect youth food and beverage intake.	There was an increase in the purchase of healthier foods and beverages of 1.4 more items per week in intervention group compared to control group. A decrease in the percentage of calories from sandwiches, sweets and desserts was observed among the 13- to 15-year-olds in the intervention group compared to the control group of the same age range. No differences were found in fruit and vegetables intake.
Vik et al., 2020 [40]	Non-randomized trial 1 academic year. School Meal Project	164	10–12 years	Free healthy school meal at lunch in the intervention group compared with normal lunch in the control group.	FFQ: validated questionnaire used in the Fruits and Vegetables Make the Marks-project.	Higher weekly intake of vegetables on sandwiches adjusted for baseline intake in intervention group compared to control group.
Bartelink et al., 2019 [38]	Longitudinal quasi-experimental trial. 2 years HPSF	1676	4–12 years	(1) Full intervention: free healthy school lunch and mid-morning snack each day + structured PA sessions after lunch. (2) Partial intervention: structured PA sessions after lunch. (3) Control	Questionnaires to assess dietary behaviors and intake filled out by parents and children. Child lunch questionnaire.	Higher intake of vegetables and dairy products and lower intake of grains and butter in the full intervention compared to control. In the partial intervention, lower intake of vegetables, dairy products and butter compared to control intervention. Decrease in time spent sedentary and increase in time spent in light PA in the full intervention compared to control intervention. Improve in children’s healthy dietary behaviors in full intervention compared to control intervention.

PSFBI: Primary School Free Breakfast Initiative; PP: Primary Prevention; PP + SP: Primary Prevention + Secondary Prevention; C: control; LA Health: Louisiana Health study; OPUS School Meal: Optimal well-being, development and health for Danish children through a healthy New Nordic Diet School Meal Study; WebDASC: Web-based Dietary Assessment Software for Children; CHANGE: Creating Healthy, Active and Nurturing Growing-Up Environments study; USDA: United States Department of Agriculture; OSNAP: Out of School Nutrition and Physical Activity study; BHCK: B’more Healthy Communities for Kids study; CIQ: Child Impact Questionnaire; BKFFQ: Block Kids 2004 Food Frequency Questionnaire; CHIRPY DRAGON: Chinese Primary School Children Physical Activity and Dietary Behavior Changes study; SFFQ: Short food frequency questionnaire; HPSF: Healthy Primary School of the Future study; PA: Physical activity; FFQ: Food frequency questionnaire.

## Data Availability

Not applicable.

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
