# Peer review of "Effectiveness of Nutritional Strategies on Improving the Quality of Diet of Children from 6 to 12 Years Old: A Systematic Review"

_nutrients, 2022, doi:10.3390/nu14020372_

Round 1
Reviewer 1 Report
Dear authors,
you are investigating a real relevant topic and good systematic reviews in this field will be very helpful for further decisions in all policies. I recommend to revise the paper concerning the following aspects:
- titel: if you focused on schools/ schoolbased interventions, you should mentioned that in the titel
- please clarify the central concepts of interest (e.g. following the PICO-scheme or comparable ones)
- what is the population of interest? ALL children between 6 and 12? (normal nurished as well as malnurished as well as obese?) or rather primary schools?
- what is the intervention of interest? nutritional strategies (on institutional level?) OR nutritional interventions (on individual level?) OR lifestyle intervention (beyond nutrition, like education, physical activities?) the wording is not consistent. If the studies are not consistent (as I suppose), you should mentioned this.
- what are the primary outcomes: children related outcomes or rather structure or process outcomes (provision of healthy food, etc.)
- in the method section I miss inclusion criteria concerning the study designs (prospective studies, controlled studies, etc)
- I miss a critial appraisal of the studies (it was not just a scoping review but a systemativ review): what about concerns about risk of bias, possible confounders
- in general a theoretical framework would be nice. What a the central assumptions.
- the result chapter contains a description of the study-characteristics, but not of the study results. The results are summarized in the discussion. Ist this structure provided?
- did you consider the various cochrane reviews about school based nutritional interventions? May be that helps for clarifying the concepts, to identify gaps in research etc.
I hope my comments are understandable and helpful. Success for the revision!
Author Response
Dear authors,
you are investigating a real relevant topic and good systematic reviews in this field will be very helpful for further decisions in all policies. I recommend to revise the paper concerning the following aspects:
The authors would like to thank the reviewer for the useful observations and suggestions performed, which we hope to have fully responded.
- title: if you focused on schools/ schoolbased interventions, you should mentioned that in the title
Schools are not referred to in the title of the review because not all the studies revised focus their intervention on schools, which has now been clarified (lines 174-179, revised version).
- please clarify the central concepts of interest (e.g. following the PICO-scheme or comparable ones)
It has been now clarified (lines 158-161, revised manuscript).
Clinical question: " In children, what type of intervention is found to be effective improving the quality of the diet or the dietary pattern compared to control?”
- Population = children between the ages of 6 and 12 years
- Intervention = all kinds of interventions, based or not on provision of foods, nutritional education or lifestyle modification)
- Compared with = control group or baseline data
- Outcome of interest = improvement of diet quality or dietary pattern
- what is the population of interest? ALL children between 6 and 12? (normal nurished as well as malnurished as well as obese?) or rather primary schools?
The population of interest is all children regardless of their nutritional status. The objective is to evaluate the effect of different interventions in children so that the entire child population is represented. As stated below, research reviewed was not based only in school interventions (line 169).
- what is the intervention of interest? nutritional strategies (on institutional level?) OR nutritional interventions (on individual level?) OR lifestyle intervention (beyond nutrition, like education, physical activities?) the wording is not consistent. If the studies are not consistent (as I suppose), you should mentioned this.
There were no restrictions on type of intervention of interest. The objective of the review is to evaluate the effect of all types of interventions in the children population from 6 to 12 years old. This fact has been now corrected as follows: “Articles involving different intervention approaches were included, both isolated nutritional interventions and those that, in addition to the nutritional component, presented other components such as nutrition education and / or physical activity in their intervention. This second type of intervention was named by the authors as lifestyle interventions” (lines 174-178, revised version).
- what are the primary outcomes: children related outcomes or rather structure or process outcomes (provision of healthy food, etc.)
Primary outcomes are children related outcomes. In a secondary way, the characteristics of the different interventions are evaluated in order to understand and analyze why some interventions are effective and others are not (lines 130-135 and 172-173, revised version).
- in the method section I miss inclusion criteria concerning the study designs (prospective studies, controlled studies, etc)
It has been clarified in the revised version. All studies included were prospective controlled trial. Lines 178-179 revised version.
- I miss a critical appraisal of the studies (it was not just a scoping review but a systematic review): what about concerns about risk of bias, possible confounders
The main risk of bias in the selected studies is that the families that usually participate in these studies are those that probably follow a healthier eating pattern. It is difficult to reach the general population. This fact has been added now (lines 345-350 and 594-615, revised manuscript).
- in general a theoretical framework would be nice. What a the central assumptions.
The central assumption of this review is that the scientific evidence on interventions that improve the nutritional quality of the child population remains limited. For this reason, the present review aims to delve into the subject and provide updated data. These has been now stated (lines 122-129, revised manuscript).
- the result chapter contains a description of the study-characteristics, but not of the study results. The results are summarized in the discussion. Is this structure provided?
Thank you for your observation. Yes, it is true that some of the results are presented in the discussion instead of in the results, but this structure has already been modified. In general, the discussion section has been extensively modified, with the aim of following the established structure as well as complying with one of the previous points, in which it was mentioned that a critical appraisal of the studies was lacking (lines 368-621, revised version).
- did you consider the various Cochrane reviews about school based nutritional interventions? May be that helps for clarifying the concepts, to identify gaps in research etc.
No other Cochrane reviews had been considered in the previous version of the manuscript. Now they have been reviewed and cited in the new one (lines 125-126, revised version).
I hope my comments are understandable and helpful. Success for the revision!
Reviewer 2 Report
Authors systematically reviewed the recent literature on eating habits intervention.
Since a correct PRISMA method was applied to include/exclude the studies considered, the most interesting point emerging from their results is that most of the included studies focus either on school or on the environment where children live, not on both situations, while all-round interventions would possibly lead to better results. I would suggest the Authors to stress this point, if they think it may be somehow important.
I don't understand why some studies have a very long duration: was the longer duration in those studied partially used for observational purposes on long-lasting effects of the intervention, or was all the time used for the intervention itself? I suggest the Authors to clarify this point.
About the same point, in the conclusion paragraph I would stress the importance, for future studies, to look at those long-lasting effects, since it is well known that the effects of interventions on eating habits of either children or adults tend to fade over time.
In spite of these minor issues, I believe that the study is well conducted, the results are correctly presented, and the conclusions agreeable.
Author Response
Authors systematically reviewed the recent literature on eating habits intervention.
- Since a correct PRISMA method was applied to include/exclude the studies considered, the most interesting point emerging from their results is that most of the included studies focus either on school or on the environment where children live, not on both situations, while all-round interventions would possibly lead to better results. I would suggest the Authors to stress this point, if they think it may be somehow important.
This is one of the points to consider and evaluate in future interventions. It is more than likely that if interventions focused on both the school and children's environment were developed there would be greater effects on the quality of the diet. And this fact has been added in lines 526-535, of the revised version.
- I don't understand why some studies have a very long duration: was the longer duration in those studied partially used for observational purposes on long-lasting effects of the intervention, or was all the time used for the intervention itself? I suggest the Authors to clarify this point.
The reviewer is right. In some studies, it was not clear if the duration of the study included only the intervention period or the entire research. This has been clarified in the revised manuscript (lines 507-513).
- About the same point, in the conclusion paragraph I would stress the importance, for future studies, to look at those long-lasting effects, since it is well known that the effects of interventions on eating habits of either children or adults tend to fade over time.
We agree with your comment, indeed this point is another important issue to keep in mind. None of the selected studies assessed long-term effects after completion of the intervention. All studies took their final measurements as soon as the intervention was completed. Thus, this has been added in the revised version (lines 514-525, revised version).
In spite of these minor issues, I believe that the study is well conducted, the results are correctly presented, and the conclusions agreeable.
We would like to thank you the reviewer for the useful comments and suggestions.